# InstructHOI: Context-Aware Instruction for Multi-Modal Reasoning in Human-Object Interaction Detection

**Jinguo Luo**[1,2], **Weihong Ren**[1][*], **Quanlong Zheng**[2][†], **Yanhao Zhang**[2][†],

**Zhenlong Yuan**[3], **Zhiyong Wang**[1], **Haonan Lu**[2], **Honghai Liu**[1]

[1]Harbin Institute of Technology, Shenzhen  [2]OPPO AI Center

[3]Institute of Computing Technology, Chinese Academy of Sciences

{23s153135, weihongren, zhiyongwang, honghai.liu}@hit.edu.cn
{zhengquanlong, zhangyanhao, luhaonan}@oppo.com
yuanzhenlong21b@ict.ac.cn

## Abstract

Recently, Large Foundation Models (LFMs), e.g., CLIP and GPT, have significantly advanced the Human-Object Interaction (HOI) detection, due to their superior generalization and transferability. Prior HOI detectors typically employ single- or multi-modal prompts to generate discriminative representations for HOIs from pretrained LFMs. However, such prompt-based approaches focus on transferring HOI-specific knowledge, but unexplore the potential reasoning capabilities of LFMs, which can provide informative context for ambiguous and open-world interaction recognition. In this paper, we propose InstructHOI, a novel method that leverages context-aware instructions to guide multi-modal reasoning for HOI detection. Specifically, to bridge knowledge gap and enhance reasoning abilities, we first perform HOI-domain fine-tuning on a pretrained multi-modal LFM, using a generated dataset with 140K interaction-reasoning image-text pairs. Then, we develop a Context-aware Instruction Generator (CIG) to guide interaction reasoning. Unlike traditional language-only instructions, CIG first mines visual interactive context at the human-object level, which is then fused with linguistic instructions, forming multi-modal reasoning guidance. Furthermore, an Interest Token Selector (ITS) is adopted to adaptively filter image tokens based on context-aware instructions, thereby aligning reasoning process with interaction regions. Extensive experiments on two public benchmarks demonstrate that our proposed method outperforms the state-of-the-art ones, under both supervised and zero-shot settings.

## 1 Introduction

Human-Object Interaction (HOI) detection plays a crucial role in high-level human-centric understanding, with applications across various domains [1, 2, 3]. The purpose of HOI detection is to detect a series of interactive triplets (i.e., ⟨*human*, *action*, *object*⟩) in open-world scenarios. This task can be specifically divided into two sub-tasks: localizing interactive human-object pairs and recognizing their interaction relationships.

Traditional HOI detectors can be primarily classified into one-stage and two-stage approaches. One-stage methods [4, 5, 6, 7] treat HOI detection as a unified multi-task learning problem, utilizing

---

[*]Corresponding Author
[†]Project Leader

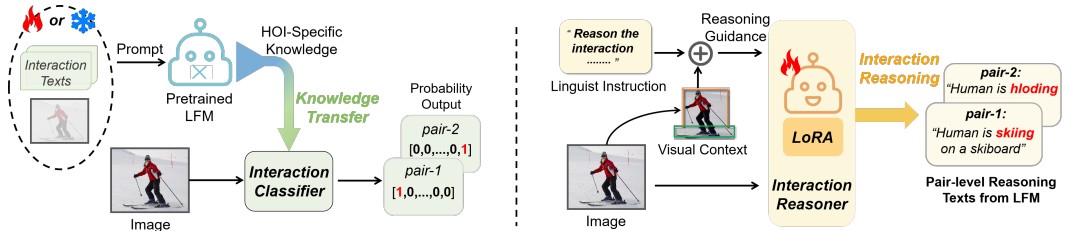

**(a)** Traditional Knowledge Transfer Methods

**(b)** Context-Aware Instruction for Interaction Reasoning

Figure 1: (a) Traditional LFM-based methods typically employ predefined or learnable prompts to transfer HOI-specific knowledge from pretrained LFM for interaction classification. (b) Our InstructHOI integrates visual interactive context with linguistic instructions to guide HOI-domain LFM in performing multi-modal interaction reasoning, generating pair-level interaction texts.

a multi-branch network to simultaneously perform human-object pair detection and interaction prediction. In contrast, two-stage ones [8, 9] first detect human-object instances using an off-the-shelf object detector, subsequently predicting interaction categories based on the visual features extracted from instance areas. Despite significant efforts in feature extraction strategies [10, 11, 12] and architecture improvements [13, 14, 15, 16], accurately identifying complex HOIs within open-world context remains challenging when relying solely on visual representation learning.

To further explore discriminative interaction representations, recent researches transfer HOI-specific knowledge from pretrained LFMs, including Vision-Language Models (VLMs) and Large Language Models (LLMs), using predefined or learnable prompts, as illustrated in Fig. 1(a). For the VLM-style methods, some earlier works [17, 18, 19] leverage static template prompts (e.g., "a photo of a person [action] a/an [object]") to derive linguistic prior knowledge from CLIP [20] at the category level. Subsequently, CMMP [21] introduces learnable multi-modal prompts, facilitating the adaptive transfer of semantic knowledge from CLIP at the instance level. Furthermore, the LLM-style methods [22, 23, 24] adopt language foundation models (e.g., ChatGPT ) to generate finer-grained descriptive texts as interactive clue prompts, thereby transferring the generalizable knowledge of LLMs for HOI detection. However, these prompt-based methods primarily focus on knowledge transfer, but fail to exploit LFMs' reasoning capabilities which can provide informative context for ambiguous and open-world interaction recognition.

According to the aforementioned challenges, we propose InstructHOI, which leverages context-aware instructions to direct LFM in performing multi-modal reasoning for HOI detection, as depicted in Fig. 1(b). Specifically, for a pretrained LFM, we first perform HOI-domain fine-tuning to bridge the inherent knowledge gap between general and HOI domains [25] and enhance its interaction reasoning capability, using a light-weight strategy, i.e., LoRA [26]. Due to the limited availability of HOI reasoning data [25], we created a large-scale dataset containing 140K image-text pairs by aggregating five existing image-only HOI datasets and transforming the one-hot labels into interaction-reasoning conversations. Then, we develop a Context-aware Instruction Generator (CIG) to guide interaction reasoning. Unlike traditional language-only instructions [27, 28], CIG first mines visual interactive context (i.e., appearance and spatial context) at the human-object level. Next, the visual context is projected into linguistic space using a two-layer instruction projector, and then is fused with linguistic instructions, providing pair-level context guidance for multi-modal interaction reasoning. Furthermore, an Interest Token Selector (ITS) is adopted to adaptively filter informative image tokens based on the context-aware instructions and reorganize the reasoning token sequences, thereby aligning the reasoning process with interaction regions.

In this paper, our motivation is to explore the potential reasoning capability of LFMs to improve HOI detection. Unlike previous LFM-based approaches, our work directly leverages tailored instructions to guide LFM in facilitating multi-modal reasoning, thereby achieving open-world interaction recognition. Besides, we enhance traditional linguistic instructions by incorporating visual interactive context at the human-object level, thus providing pair-level multi-modal reasoning guidance. To summarize, our contributions are as follows:

- For a pretrained LFM, to bridge the gap between general and HOI-domain knowledge, we build a high-quality interaction-reasoning dataset and perform supervised fine-tuning using a lightweight strategy.

- We develop a Context-aware Instruction Generator (CIG) to enhance linguistic instructions by incorporating informative visual context at the human-object level, providing multi-modal reasoning guidance.

- To align the reasoning process with interaction regions, an Interest Token Selector (ITS) is adopted to adaptively filter and reorganize reasoning token sequences based on context-aware instructions.

- We evaluate our InstructHOI on two benchmarks: HICO-DET and V-COCO, and it outperforms the state-of-the-art methods, achieving superior performance in both supervised and zero-shot settings.

## 2   Related Work

**Traditional HOI Detectors:**  Traditional HOI detectors can be primarily classified into one-stage and two-stage approaches. One-stage methods regard HOI detection as a multi-task learning, aiming to simultaneously perform object detection and interaction prediction. Earlier methods [29, 5, 4] typically adopt a multi-branch CNN architecture for parallel human-object instance localization and interaction recognition. Then, some auxiliary priors (e.g., interaction points [4] and union boxes [30]) are introduced to align instances with their corresponding interactions. Recently, Transformer-based methods [7, 6, 31] take a prominent position, due to their exceptional context capture ability. However, such a disentangled architecture may suffer from insufficient context exchange between the branches, leading to inferior prediction performance.

Two-stage methods treat HOI detection as two sequential sub-tasks. They initially localize human-object instances with an off-the-shelf detector and then identify interactions leveraging the visual features extracted from the instance regions. The early CNN-based methods [32, 33, 34, 35, 36, 9] strive to extract rich visual interaction representations, e.g., spatial relationship [37], gaze attention [11] and pose feature [33] to assist HOI detection. Recent Transformer-based methods [38, 39, 40, 41, 42] attempt to improve the vanilla Transformer for enhancing feature extraction of HOI detection. Despite significant efforts in feature extraction strategies and architecture enhancements, accurately distinguishing complex HOIs in open world remains challenging when relying solely on visual representation learning.

**LFM-based HOI Detectors:**  LFM-based HOI detectors can be primarily classified into VLM-style and LLM-style. To further explore discriminative HOI representations, recent approaches [43, 44, 45] seek to extract prior knowledge from VLMs [46, 47] by leveraging their distinctive ability to unify visual and linguistic features. Among VLM-style methods, the pioneering works [44, 19] typically transform one-hot labels into annotation texts via a static prompt template, e.g., "a photo of a person [action] a/an [object]". These annotations are then encoded as linguistic priors using CLIP, enabling category-level knowledge transfer. In addition, MP-HOI [43] utilizes extra visual prompts to provide fine-grained visual priors, and aims to eliminate the ambiguity in linguistic descriptions. Furthermore, CMMP [21] introduces learnable multi-modal prompts, facilitating the adaptive transfer of semantic knowledge from CLIP at the instance level.

LLM-style methods [22, 23, 24, 48] usually employ language foundation models to generate finer-grained descriptive texts as interactive prompts, which can transfer the generalizable knowledge from LLMs to improve HOI detection. E.g., UniHOI [22] designs a knowledge retrieval process for ChatGPT to acquire comprehensive explanations for each HOI category, which provides rich contextual information for interaction prediction. CMD-SE [24] introduces a two-step GPT-querying mechanism to produce descriptions of human body, and thus generate general body-part prompts, which is helpful for recognizing ambiguous actions. However, the existing LFM-based methods primarily focus on transferring HOI-specific knowledge, but fail to explore the reasoning capabilities of LFMs, leading to incomplete exploration of their full potential, particularly for open-world interaction recognition.

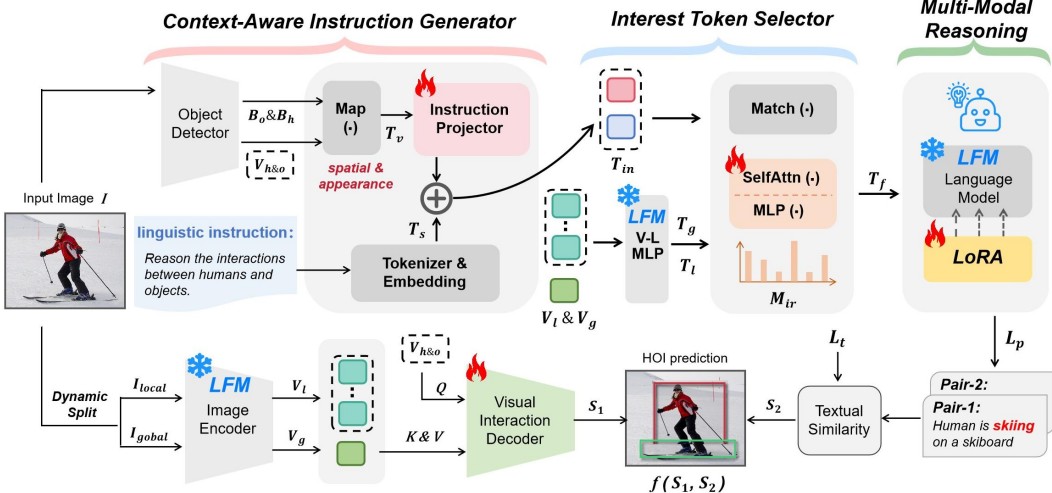

Figure 2: InstructHOI involves two interaction prediction branches: the multi-modal reasoning branch and the visual interaction decoder. The former includes Context-aware Instruction Generator (CIG), Interest Token Selector (ITS), and Multi-Modal Reasoning (MMR). CIG combines visual context with linguistic instructions to generate context-aware instructions $T_{in}$ (Sec. 3.3). ITS then adaptively filters image tokens and reorganizes the reasoning token sequences $T_f$ to align the reasoning process with interaction regions (Sec. 3.4). Finally, the LFM's language model in MMR, fine-tuned with LoRA, uses $T_f$ to conduct multi-modal reasoning, generating pair-level interaction texts (Sec. 3.5). Meanwhile, the visual interaction decoder utilizes pair and global image features to perform interaction decoding (Sec. 3.5).

# 3 Method

## 3.1 Overall Architecture

The overall architecture of InstructHOI is illustrated in Fig. 2. Given an image $I$, an off-the-shelf object detector (i.e., DETR [49]) is first employed to localize human and object instances ($B_h$, $B_o$), and then obtain the Human-Object (H-O) pair features $V_{h\&o}$ by concatenating the instance features from DETR for each H-O pair. Meanwhile, following the dynamic image encoding strategy in [50], we dynamically split the image $I$ and obtain the global and local images ($I_{global}$, $I_{local}$), which are then separately encoded into the global and local image features ($V_g$, $V_l$), using the pretrained image encoder of LFM (i.e., InternVL2 [50]).

InstructHOI involves two interaction prediction branches: the multi-modal reasoning branch and the visual interaction decoder. The former branch mainly includes three components: Context-aware Instruction Generator (CIG), Interest Token Selector (ITS), and Multi-Modal Reasoning (MMR). To direct LFM in facilitating multi-modal reasoning and achieving pair-level interaction prediction, CIG first extracts the appearance and spatial context embedding $T_v$ of each H-O pair, which is then inserted into linguistic instructions, forming pair-specific context-aware instructions $T_{in}$ (Sec. 3.3). Furthermore, to align reasoning process with interaction regions, ITS filters informative tokens from local image tokens $T_l$ based on the instructions $T_{in}$, and then reorganizes them into reasoning token sequences $T_f$ (Sec. 3.4). The LFM's language model in MMR, fine-tuned with **LoRA**, utilizes the filtered token sequences $T_f$ to achieve pair-level interaction reasoning and acquire interaction-reasoning probability distribution $S_2$ (Sec. 3.5). Meanwhile, in the visual interaction decoder, the pair features $V_{h\&o}$ act as *Query*, while the global image features $V_g$ act as *Key* and *Value*, performing interaction decoding and yielding interaction-decoding probability distribution $S_1$ (Sec. 3.5). Finally, both distributions $S_1$ and $S_2$ are combined to yield the final interaction score.

## 3.2 HOI-domain Fine-tuning

Different from task-specific models, Large Foundation Models (LFMs) are typically pretrained on vast and diverse datasets, acquiring general-domain knowledge across both visual and linguistic modalities. However, such general models often struggle to achieve accurate zero-shot interaction prediction, due to the gap between general knowledge and that specific to HOI domain [25]. To bridge the knowledge gap and enhance the interaction reasoning capability for HOI detection, we

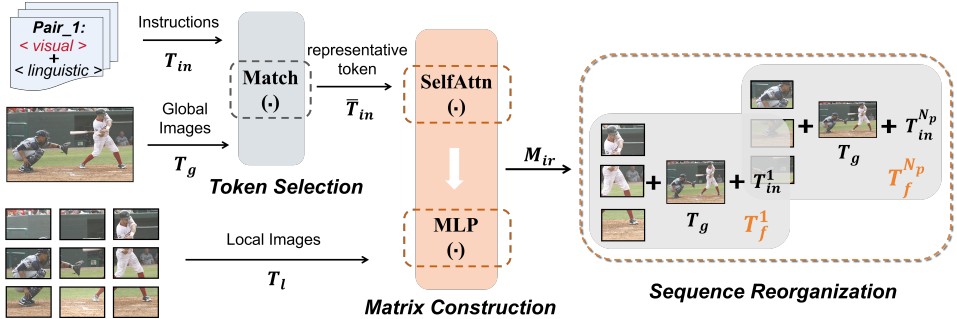

Figure 3: The illustration of Interest Token Selector. It contains three steps: token selection, matrix construction and sequence reorganization.

introduce a multi-modal LFM (i.e., InternVL2) and conduct HOI-domain fine-tuning on a generated high-quality dataset.

Specifically, we employ a light-weight adaptation strategy (i.e., LoRA) to facilitate efficient and low-consumption fine-tuning. As shown in Fig. 2, we freeze the entire pretrained LFM model, including the image encoder, the Vision-Language MLP (V-L MLP), and the language model, while only training a few injectable parameters (about **0.8%** of the pretrained model's parameters) for the language model of InternVL2, acquiring fine-tuned model $\pi_\theta^{lora}$. Additionally, due to the limited availability of HOI reasoning data [25], we aggregate five existing image-only HOI datasets [51, 52, 53, 54, 55] and build a high-quality dataset containing **140K** image-text pairs across 1K object categories, 600 action categories, and 15K HOI categories. To acquire interaction reasoning texts, we transform original one-hot labels into *Question&Answer* conversations. E.g., for a human-bike pair with 'sit' and 'ride' interactions, the conversation is formulated as: {*Question: Reason the interaction relationships between humans and objects in the image; Answer: Human is sitting on and riding a bike*}. Moreover, to avoid potential data leakage during evaluation, all testing data used in the experiments are excluded from the fine-tuning dataset. The visualization of the dataset is presented in the Supplementary Material.

### 3.3 Context-aware Instruction Generator (CIG)

Recent studies [56, 57] explore the potential reasoning capabilities of LFMs for tackling complex visual tasks using different strategies, e.g., chain-of-thought language instruction [27]. However, reasoning about the ambiguous interactions in complex scenarios needs accurate guidance to assist LFM in understanding the spatial relationship and instance appearance of each H-O pair, while language-only instructions can hardly provide these visual context clues. To remedy this, we develop a Context-aware Instruction Generator (CIG), which incorporates the informative visual context of each H-O pair in linguistic instructions, providing multi-modal guidance for pair-level interaction reasoning.

As shown in Fig. 2, we first derive the pair features $V_{h\&o}$ by combining the instance features for each H-O pair, which are extracted from the hierarchical backbone of the object detector and contain discriminative characteristics of each instance. Then, the multi-level $V_{h\&o}$ are flattened and concatenated, serving as the visual appearance representations of each H-O pair. To further encourage LFM to be aware of spatial relationships, we encode H-O spatial context based on the localization boxes $B_h\&B_o$. Following the previous works [58, 21], we extract spatial features from $B_h\&B_o$, by calculating the intersection-over-union, scaled distance, spatial direction, etc. Subsequently, both appearance and spatial features are mapped to a dimensionality of $d$, forming the visual context embedding $T_v$. The process can be formulated as:

$$T_v = \mathrm{Map}\left(\mathrm{SPEnc}\left(B_h, B_o\right), V_{h\&o}\right), \tag{1}$$

where $\mathrm{SPEnc}(\cdot)$ represents spatial feature encoding, and $\mathrm{Map}(\cdot)$ indicates feature mapping operation. Meanwhile, the linguistic instructions are also encoded into linguistic context embedding $T_s$ using the tokenizer of InternVL2.

Inspired by the "ViT $\rightarrow$ V-L MLP $\rightarrow$ LLM" architecture in the existing studies (e.g., LLaVA [59]), we leverage a two-layer instruction projector to map the visual context embedding $T_v$ into the linguistic embedding space, eliminating the knowledge gap between visual and linguistic modalities. Finally, the pair-level visual context embedding $T_v$ is fused with the linguistic context embedding $T_s$, forming

pair-level context-aware instructions $T_{in}$. Overall, the process can be formulated as:

$$T_{in} = \text{Concat}\left(\text{Proj}_I\left(T_v\right), T_s\right), \tag{2}$$

where $\text{Proj}_I(\cdot)$ indicates the instruction projector, which consists of a two-layer MLP followed by a GeLU layer and $\text{Concat}(\cdot)$ denotes the concatenation operation.

### 3.4 Interest Token Selector (ITS)

According to the dynamic image encoding strategy [50], input image $I$ is typically dynamically split, acquiring global and local images ($I_{global}$, $I_{local}$), where the global image $I_{global}$ provides a holistic image context, and the local images $I_{local}$ offer region-level context information. Considering that, within an image, the interactive regions of H-O pairs also dynamically vary based on their locations and interaction categories. E.g., for ⟨*human*, *hold*, *apple*⟩, the interactive region mainly focuses on the hand regions, while for ⟨*human*, *kick*, *football*⟩, the interactive region shifts to the leg regions. To align pair-level reasoning with the corresponding interaction regions, we develop an Interest Token Selector (ITS), which evaluates the interaction relevance of local images $I_{local}$ for each H-O pair, based on the context-aware instructions $T_{in}$. Thus, ITS can adaptively select the informative image tokens and then reorganizes the reasoning token sequence for each H-O pair.

Following the multi-modal reasoning mechanism in InternVL2, the image features ($V_l$, $V_g$) are projected into linguistic space using the V-L MLP of InternVL2, acquiring image tokens ($T_l$, $T_g$). Additionally, as depicted in Fig. 3, the Interest Token Selector contains three steps: token selection, matrix construction and sequence reorganization. Firstly, to extract the representative tokens from the instructions $T_{in}$, we calculate the cosine similarity between $T_{in}$ and global image tokens $T_g$, and select the *top-n* most similar ones as representative instruction tokens $\overline{T}_{in}$, as follows:

$$\overline{T}_{in} = \text{Match}\left(T_{in}, T_g\right) \in \mathbb{R}^{N_p \times n \times d}, \tag{3}$$

where $\text{Match}(\cdot)$ represents the cosine similarity operation and selection, $N_p$ and $d$ indicate the number of H-O pairs and token dimension, respectively. Afterwards, a self-attention layer is employed to facilitate feature fusion and context propagation between $\overline{T}_{in}$ and the local image tokens $T_l$. An MLP is then applied to predict the interaction relevance of local images for each H-O pair, constructing interaction-relevance matrix $M_{ir}$, as follows:

$$M_{ir} = \text{MLP}(\text{SelfAttn}(Q, K, V : \text{concat}(\overline{T}_{in}, T_l))) \in \mathbb{R}^{N_p \times N_l}, \tag{4}$$

where $\text{SelfAttn}(\cdot)$ represents a self-attention layer and $N_l$ indicates the number of local images.

Finally, we use softmax operation to calculate the relevance probability distribution based on the matrix $M_{ir}$, subsequently selecting the informative tokens from $T_l$ and reorganizing the reasoning token sequence $T_f$ for each H-O pair in the format of [⟨*selected local images tokens*⟩, ⟨*global image tokens*⟩, ⟨*instruction tokens*⟩], as follows:

$$T_f = \text{Concat}(\widetilde{T}_l, T_g, T_{in}), \quad \text{where} \ \ \widetilde{T}_l = \text{Filter}\left(T_l, M_{ir}\right), \tag{5}$$

$\widetilde{T}_l$ represents the selected local images tokens and $\text{Filter}(\cdot)$ indicates the image token filtering operation based on the interaction-relevance matrix. The visualization of ITS are provided in the Supplementary Material.

### 3.5 Inference and Training

**Inference.** The inference process is illustrated in Fig. 2. InstructHOI involves two interaction prediction branches: the visual interaction decoder and the multi-modal reasoning branch. In the visual interaction decoder, the interaction decoding is performed based on the visual representations, taking pair features $V_{h\&o}$ as *Query* and global image features $V_g$ as *Key* and *Value*, and yields interaction-decoding probability distribution $S_1$:

$$S_1 = \text{Proj}_d(\text{CrossAttn}(Q : V_{h\&o}; K, V : V_g)) \in \mathbb{R}^{N_p \times N_c}, \tag{6}$$

where $\text{CrossAttn}(\cdot)$ indicates the cross-attention operation, and $\text{Proj}_d(\cdot)$ represents the distribution projector, which consists of a MLP followed by a sigmoid operation, and $N_c$ represents the number of interaction categories.

For the multi-modal reasoning branch, the fine-tuned InternVL2, $\boldsymbol{\pi}_{\boldsymbol{\theta}}^{lora}$, utilizes the refined reasoning token sequence $T_f$ to conduct multi-modal reasoning and generate pair-level interaction text descriptions. Next, we calculate textual cosine similarity between $L_p$ and textual HOI labels $L_t$, and then compute interaction-reasoning probability distribution $S_2$, following [43]:

$$S_2 = \text{Softmax}\left(\text{F}_{\cos}\left(L_p, L_t\right)\right), \quad \text{where} \quad L_p = \boldsymbol{\pi}_{\boldsymbol{\theta}}^{lora}\left(T_f\right) \tag{7}$$

where $\text{F}_{\cos}(\cdot)$ indicates the cosine similarity operation, and $S_2 \in \mathbb{R}^{N_p \times N_c}$. Finally, the total interaction prediction score is obtained by combining distributions $S_1$ and $S_2$:

$$S_{hoi} = (S_h)^{\lambda} \cdot (S_o)^{\lambda} \cdot S_1 \cdot S_2, \tag{8}$$

where $S_h$ and $S_o$ indicate the detection scores of human and object instances from the object detector, respectively.

**Training.** To supervise the visual interaction decoder, we employ the following Focal Loss:

$$\mathcal{L}_v = \frac{1}{\sum_{i=1}^{N_p}\sum_{c=1}^{N_c}\mathbf{y}^{i,c}} \sum_{i=1}^{N_p}\sum_{c=1}^{N_c} \text{FocalLoss}(\mathbf{y}^{i,c}, S_1^{i,c}), \tag{9}$$

where $\mathbf{y}^{i,c} \in \{0, 1\}$ in $\mathbf{y}$ indicates whether the groundtruth of the $i$-th human-object pair contains the $c$-th interaction class and $S_1^{i,c}$ in $S_1$ is the corresponding predicted probability. In addition, to supervise the multi-modal reasoning branch, we adopt similarity constraint loss, as follows:

$$\mathcal{L}_{sim} = -\frac{1}{N_p}\sum_{i=1}^{N_p} \log \frac{\sum_{c=1}^{N_c}\mathbf{y}^{i,c} \cdot \mathbf{Z}(i,c)}{\sum_{j=1}^{N_c}\mathbf{Z}(i,j)}, \quad \text{where} \quad \mathbf{Z}(i,j) = \exp(\text{F}_{\cos}(L_p^i, L_t^j)). \tag{10}$$

Overall, the total loss function is formulated as: $\mathcal{L} = \mathcal{L}_v + \alpha\mathcal{L}_{sim}$.

# 4 Experiments

## 4.1 Experimental Setting

**Datasets.** Following previous works, we conduct experiments on two commonly used HOI datasets: HICO-DET [51] and V-COCO [52]. The HICO-DET dataset comprises 47776 images, with 38118 for training and 9658 for testing, covering 117 actions, 80 objects, and 600 HOIs. Additionally, the 600 HOIs are divided into 138 Rare and 462 Non-Rare categories based on the sample distribution. The V-COCO dataset, contains 10346 images, including 5400 in the trainval set, and 4946 in the test set, across 29 actions, 80 objects, and 259 HOIs.

**Evaluation Metric.** Following the standard metric, the mean Average Precision (mAP) is adopted to evaluate the performance of InstructHOI. During the evaluation, a true positive HOI triplet needs to meet two criteria: 1) the predicted human and object bounding boxes should have Intersection over Union (IoU) values greater than 0.5 with the ground truth, and 2) the HOI classification is correct.

**Implementation Details.** We take the DETR for object detection and adopt the pretrained InternVL2$_{1b}$ as the foundation model. During training, we freeze the external models (DETR and InternVL2) and update the parameters of the remaining components in InstructHOI. The entire InstructHOI model is trained on four Tesla A800 GPUs with a batch size of 16 for 20 epochs, using the AdamW [60] optimizer.

## 4.2 Comparisons with the State-of-the-Arts

### 4.2.1 Supervised Setting

In Table 1, we present the quantitative results for the supervised setting on the HICO-DET and V-COCO datasets, respectively. Notably, our method outperforms all the existing state-of-the-art methods on both datasets. For the HICO-DET dataset, InstructHOI achieves remarkable mAPs of **47.68** and **49.89** in the default and known object full settings. Compared to recent state-of-the-art HOI detectors RLIPv2 [61] and Pose-Aware [62], our model obtains significant performance gains of 2.59 mAP (relatively 5.74%) over RLIPv2 and 1.67 mAP (relatively 3.63%) over Pose-Aware,

Table 1: Performance comparison on HICO-DET and V-COCO datasets. For results on HICO-DET, we follow commonly used experimental setting to fine-tune the object detector on its training set.

| Method | Backbone | HICO-DET | | | | | | V-COCO | |
| | | Default | | | Known Object | | | | |
| | | Full | Rare | Non-Rare | Full | Rare | Non-Rare | $AP_{role}^{\#1}$ | $AP_{role}^{\#2}$ |
|---|---|---|---|---|---|---|---|---|---|
| *One-stage methods* | | | | | | | | | |
| FGAHOI [63] | Swin-L | 37.18 | 30.71 | 39.11 | 38.93 | 31.93 | 41.02 | - | - |
| RLIPv2 [61] | Swin-L | 45.09 | 43.23 | 45.64 | - | - | - | 72.1 | 74.1 |
| *Two-stage methods* | | | | | | | | | |
| PViC [64] | Swin-L | 44.32 | 44.61 | 44.24 | 47.81 | 48.38 | 47.64 | 64.1 | 70.2 |
| Pose-Aware [62] | Swin-L | 46.01 | 46.74 | 45.80 | 49.50 | 50.59 | 49.18 | 63.0 | 68.7 |
| *LFM-based methods* | | | | | | | | | |
| EZ-HOI [48] | R50+ViT-L | 38.61 | 37.70 | 38.89 | - | - | - | 60.5 | 66.2 |
| UniHOI-l [22] | R101+ViT-L | 40.95 | 40.27 | 41.32 | 43.26 | 43.12 | 43.25 | 68.1 | 70.8 |
| DiffusionHOI [65] | ViT-L | 42.54 | 42.95 | 42.35 | 44.91 | 45.18 | 44.83 | 67.1 | 71.1 |
| MP-HOI [43] | Swin-L+ViT | 44.53 | 44.48 | 44.55 | - | - | - | 66.2 | 67.6 |
| SICHOI [23] | R101+ViT-L | 45.04 | 45.61 | 44.88 | 48.16 | 48.37 | 48.09 | 71.1 | 75.6 |
| InstructHOI (Ours) | R50+ViT-L | 45.95 | 46.51 | 45.78 | 48.57 | 49.23 | 48.37 | 70.8 | 74.2 |
| InstructHOI (Ours) | R101+ViT-L | **47.68** | **47.97** | **47.59** | **49.89** | **50.92** | **49.58** | **72.4** | **76.1** |

respectively. By introducing HOI-domain LFM, InstructHOI can extract rich contextual clues and discriminative interaction representations, to tackle ambiguous interaction detection in complex scenarios. Additionally, compared to the state-of-the-art LFM-based methods SICHOI [23] and MP-HOI [43], InstructHOI also achieves significant performance improvements, outperforming SICHOI by 2.64 mAP (relatively 5.86%) and MP-HOI by 3.15 mAP (relatively 7.07%), respectively, in the commonly used default full setting. Specifically, VLM-style methods (e.g., ADA-CM and MP-HOI) adopt single- or multi-modal prompts to transfer HOI-specific knowledge from LFM, while LLM-style approaches (e.g., UniHOI and SICHOI) generate comprehensive descriptions as interactive prompt based on language foundation models. However, all these LFM-based methods primarily focus on transferring HOI-specific knowledge, without exploring the potential reasoning capabilities of LFMs. Unlike existing knowledge transfer methods, our InstructHOI directly leverages context-aware instructions to guide LFM in facilitating pair-level multi-modal reasoning, acquiring discriminative interaction representations for ambiguous and open-world interaction recognition.

For V-COCO dataset, as reported in the right part of Table 1, InstructHOI also performs the best among all the state-of-the-art methods, achieving $AP_{role}^{\#1}$ of **72.4** in scenario #1 and $AP_{role}^{\#2}$ of **76.1** in scenario #2. Specifically, comparing to the recent HOI detectors RLIPv2 and SICHOI, InstructHOI demonstrates superior performance, e.g., $AP_{role}^{\#1}$ of 72.4 vs 72.1 and 71.1, and $AP_{role}^{\#2}$ of 76.1 vs 74.1 and 75.6. Even using "Resnet50" backbone, our method still performs better than most of the state-of-the-art approaches. The superiority of our proposed InstructHOI comes from the fact that we fully exploit the reasoning ability of large foundation models rather than simply transferring knowledge.

### 4.2.2 Zero-shot Setting

Consistent with previous zero-shot experiments [17, 66, 22], we evaluate our method on HICO-DET under four zero-shot settings: 1) Rare First Unseen Combination (RF-UC) constructs training set with all the object and verb categories but excludes a certain number of rare HOI categories. 2) Non-rare First Unseen Combination (NF-UC) prioritizes non-rare interactions when selecting the held-out HOI categories. 3) Unseen Object (UO) is designed to assess interaction recognition with novel object categories. 4) Unseen Verb (UV) focuses on discovering novel action categories. For a fair comparison, we present recent LFM-based zero-shot HOI detectors with the same "ResNet50" backbone in Table 2, where our InstructHOI surpasses all other methods across four zero-shot settings.

For RF-UC and NF-UC settings, InstructHOI achieves **36.82** mAP and **36.42** mAP for unseen HOI categories, respectively. Compared to the latest method SICHOI, our approach achieves gains of 3.14 mAP and 2.58 mAP in the RF-UC full and unseen settings, respectively, as well as gains of 2.59 mAP and 1.90 mAP in the NF-UC full and unseen settings, respectively. The reason is that our proposed InstructHOI has interactive reasoning capabilities, rather than simply borrowing general knowledge from the LFMs. As for UO and UV settings, InstructHOI attains mAPs of **39.92** and **31.64** for unseen HOI categories, respectively. Compared to the latest method CMMP, our approach achieves improvements of 0.79 mAP and 0.25 mAP in the UO full and unseen settings, respectively,

Table 2: Zero-shot generalization on HICO-DET [51].

| Methods | Type | Full | Seen | Unseen |
|---|---|---|---|---|
| DiffusionHOI [65] | RF-UC | 35.89 | 36.77 | 32.06 |
| EZ-HOI [48] | RF-UC | 36.73 | 37.35 | 34.24 |
| CMMP [21] | RF-UC | 37.13 | 37.42 | 35.98 |
| SICHOI [23] | RF-UC | 40.11 | 41.58 | 34.24 |
| InstructHOI | RF-UC | **43.25** | **44.86** | **36.82** |
| BCOM [67] | NF-UC | 32.03 | 31.76 | 33.12 |
| HOIGen [68] | NF-UC | 33.08 | 32.86 | 33.98 |
| CMMP [21] | NF-UC | 35.13 | 35.53 | 33.52 |
| SICHOI [23] | NF-UC | 35.75 | 36.06 | 34.52 |
| InstructHOI | NF-UC | **38.34** | **38.82** | **36.42** |
| UniHOI [22] | UO | 31.56 | 34.76 | 19.72 |
| HOIGen [68] | UO | 33.48 | 32.90 | 36.35 |
| EZ-HOI [48] | UO | 36.38 | 36.02 | 38.17 |
| CMMP [21] | UO | 36.74 | 36.15 | 39.67 |
| InstructHOI | UO | **37.53** | **37.05** | **39.92** |
| HOIGen [68] | UV | 32.34 | 34.31 | 20.27 |
| UniHOI [22] | UV | 34.68 | 36.78 | 26.05 |
| CMMP [21] | UV | 36.38 | 37.28 | 30.84 |
| EZ-HOI [48] | UV | 36.84 | 38.15 | 28.82 |
| InstructHOI | UV | **38.12** | **39.17** | **31.64** |

Table 3: Performance contribution of each component.

| Method | HICO-DET (Default) | | | V-COCO | |
|---|---|---|---|---|---|
| | Full | Rare | Non-Rare | $AP_{role}^{\#1}$ | $AP_{role}^{\#2}$ |
| Base | 36.21 | 32.84 | 37.22 | 64.8 | 70.1 |
| +MMR | 43.06 | 43.40 | 42.96 | 68.9 | 72.4 |
| +MMR+CIG | 44.98 | 44.69 | 45.07 | 70.2 | 73.9 |
| +MMR+CIG+ITS | **45.95** | **46.51** | **45.78** | **70.8** | **74.2** |

Table 4: Effect of Context-aware Instruction Generator.

| Method | HICO-DET (Default) | | | V-COCO | |
|---|---|---|---|---|---|
| | Full | Rare | Non-Rare | $AP_{role}^{\#1}$ | $AP_{role}^{\#2}$ |
| Base+MMR | 43.06 | 43.40 | 42.96 | 68.9 | 72.4 |
| +SC | 43.84 | 43.87 | 43.83 | 69.3 | 72.9 |
| +AC | 44.43 | 44.10 | 44.53 | 69.8 | 73.6 |
| +AC+SC | **44.98** | **44.69** | **45.07** | **70.2** | **73.9** |

Table 5: Effect of Interest Token Selector.

| Image Token | HICO-DET (Default) | | | V-COCO | |
|---|---|---|---|---|---|
| | Full | Rare | Non-Rare | $AP_{role}^{\#1}$ | $AP_{role}^{\#2}$ |
| $T_g$ | 43.62 | 42.29 | 44.02 | 69.3 | 72.7 |
| $T_g + T_l$ | 44.98 | 44.69 | 45.07 | 70.2 | 73.9 |
| $T_g + \widetilde{T}_l$ | **45.95** | **46.51** | **45.78** | **70.8** | **74.2** |

and 1.74 mAP and 0.80 mAP in the UV full and unseen settings, respectively. All the four zero-shot experimental results consistently demonstrate the effectiveness of our InstructHOI in detecting unseen and novel HOIs. By leveraging the interaction reasoning capabilities of HOI-domain LFMs, InstructHOI exhibits superior open-world interaction detection performance and generalization.

## 4.3 Ablation Study

In this subsection, we evaluate the effects of Multi-Modal Reasoning (MMR), Context-aware Instruction Generator (CIG), and Interest Token Selector (ITS) components in InstructHOI on both the HICO-DET and V-COCO datasets. For a fair comparison, we create a baseline mode (denoted as "Base") by simply combining the DETR (using Resnet50 as backbone) and visual interaction decoder branch (using ViT-L as backbone), which represents a degraded version of InstructHOI without MMR, CIG, and ITS. Here, the standalone 'MMR' refers to multi-modal reasoning with language-only instructions, while 'MMR+CIG' denotes multi-modal reasoning with context-aware instructions (see subsection 3.3). Additional ablation studies on advanced object detector, HOI-domain fine-tuning, and the number of representative tokens are provided in the Supplementary Material.

**Component Ablation.**   As shown in Table 3, each component of InstructHOI significantly enhances the baseline model. Specifically, MMR improves the baseline by 6.85 mAP in full setting on HICO-DET, while CIG provides an additional improvement of 1.92 mAP. Ultimately, the combination of all the three components results in a total improvement of 9.74 mAP. The above results demonstrate that the reasoning capabilities of LFM can significantly improve HOI detection, as well as highlight the effectiveness of CIG and ITS in further improving LFM's reasoning abilities.

**Context-aware Instruction Generator.**   Within the Context-aware Instruction Generator (CIG), we integrate visual context into linguistic instructions to enhance the spatial and appearance understanding of LFM. In Table 4, we evaluate the Spatial Context (SC) and the Appearance Context (AC) in CIG separately, based on the "Base + MMR" model (i.e., reasoning with language-only instructions). The results demonstrate that both AC and SC can enhance the language-only instructions, providing context guidance for interaction reasoning.

**Interest Token Selector.**   As shown in Table 5, we evaluate the effectiveness of the Interest Token Selector (ITS) by using different combinations of image tokens: $T_g$ (global image tokens), $T_l$ (local image tokens), and $\widetilde{T}_l$ (selected local image tokens). Specifically, the combination of $(T_g + \widetilde{T}_l)$ outperforms $(T_g + T_l)$ by 0.97 mAP in the full setting of HICO-DET. This indicates that the ITS effectively filters informative image tokens from $T_l$, aligning the reasoning with interaction areas.

# 5 Conclusion

In this paper, we propose a novel LFM-based HOI detector, InstructHOI. Different from the existing LFM-based approaches, InstructHOI directly learns tailored instructions to guide LFM in facilitating multi-modal reasoning, and thus can improve the open-world interaction recognition. Specifically, we develop a Context-aware Instruction Generator (CIG) to enhance linguistic instructions by incorporating visual interactive context, forming pair-level reasoning guidance. Furthermore, an Interest Token Selector (ITS) is adopted to align reasoning process with interaction regions. Extensive experiments on two public benchmarks demonstrate that our proposed method outperforms the state-of-the-art ones, under both supervised and zero-shot settings. Ablation studies also prove the effectiveness of each component in our proposed InstructHOI.

## Acknowledgment

This work was supported in part by the National Natural Science Foundation of China under Grants 62206075, 62573163, 62503139, and 62261160652, in part by the GuangDong Basic and Applied Basic Research Foundation under Grant 2024A1515012028, in part by the Shenzhen Science and Technology Program under Grant GXWD20231129125006001, in part by the Science and Technology Development Fund (FDCT), Macau SAR, under Grant 0095/2022/AFJ.

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
