# OpenReview forum: "InstructHOI: Context-Aware Instruction for Multi-Modal Reasoning in Human-Object Interaction Detection"
_NeurIPS.cc/2025/Conference — NeurIPS 2025 spotlight_

### Official Review · Reviewer_jzDg · 2025-06-30

**Clarity:** 3
**Significance:** 3
**Originality:** 3
**Rating:** 4
**Confidence:** 3

**Summary:**

This paper introduces a human object interaction detection framework that utilizes the large foundation models to enhance multi-modal reasoning for HOI detection. The authors propose to fine-tune the LFM on large-scale HOI-domain data to bridge the gap between pretrained knowledge and HOI detection. The authors propose a Context-aware Instruction Generator to guide interaction reasoning and an Interest Token Selector to select important image tokens. The experimental results on two HOI benchmarks demonstrate the effectiveness of the proposed methods.

**Questions:**

- I would expect an ablation study on the HOI-domain fine-tuning data choice, such as only fine-tuning the LFM on HICO-DET. For the LFM choice, the authors could provide an ablation study on this issue or explain the rationale behind the choice of InternVL2.
- Following the previous question, could the authors clarify that the *Backbone* in Table 1 refers to which part of the proposed method?
- Could authors provide an inference and training cost comparison with baselines?

**Ethical Concerns:**

["NO or VERY MINOR ethics concerns only"]

**Final Justification:**

After reading the rebuttal and other reviewers' comments, my concerns are addressed. So I will maintain my positive score.

**Limitations:**

More ablation is expected to verify the core contribution of this work.

**Quality:**

3

**Strengths And Weaknesses:**

# Strengths
- This paper is well-written and easy to follow.
- The proposed framework is well-motivated and reasonable.
- The performance is good on popular HOI benchmarks.

# Weaknesses
- Major performance improvements come from the Multi-Modal Reasoning, which includes training on data from five HOI datasets. My concern is whether the performance gains come from additional training data or a stronger multi-modal LFM (InternVL2). These may cause unfair comparison with other baselines.
- The necessity of Spatial Context in CIG. From the results reported in Table 4, the gains from SC are quite limited, and the design of SC requires an offline object detector model, which may introduce additional noise.
- The frameworks introduce a lot of additional modules compared with the baselines. A comparison of inference and training cost is expected to reveal the trade-off between efficiency and effectiveness of proposed methods.

---

> ### Author Rebuttal · Authors · 2025-07-31
>
> Dear reviewer jzDg,
>
> We gratefully acknowledge your comprehensive review, insightful critiques, and affirming evaluation of our approach, e.g.,  "well-motivated", "reasonable" and "easy to follow".
>
> Below, we provide a detailed response to each of the points you raised (note: the metrics for HICO-DET in the following table are reported under the "default full setting", while those for V-COCO are  $AP_{role}^{\\#1}$; the references maintain their original numbering from the main paper):
>
> --------------------------------------------------------------------------------------------------------------
>
> **Q1:  Concern about additional training data or a stronger multi-modal LFM (InternVL2).**
>
> **A1:** **For finetuning data**, we have reported the results of ablation study in the supplementary materials (Table 2). As shown in the following table, the“pretrained" indicates pretrained LFM without fine-tuning; “fine-tune$^{†}$" indicates fine-tuning on a specific training dataset (i.e., HICO-DET or V-COCO); “fine-tune$^{‡}$" indicates fine-tuning on the collected dataset. Notably, even when fine-tuned exclusively on HICO-DET or V-COCO (fine-tune$^{†}$), our model still achieves SOTA performance. Moreover, the “fine-tune$^{‡}$” strategy yields an additional gain of 0.59 mAP on the HICO-DET benchmark and 0.40 mAP on the V-COCO, demonstrating that finetuning on the larger dataset could further enhance InstructHOI's generalization ability.
>
> | Strategy     | HICO-DET | V-COCO |
> |--------------|:----------:|:--------:|
> | pretrained   | 43.68    | 69.4   |
> |fine-tune$^{†}$   | 45.36    | 70.4   |
> | fine-tune$^{‡}$   | **45.95**    | **70.8**   |
>
>
> **For LFM choice**, we have reported the results of ablation study on LFMs in the supplementary materials (Table 1). As shown in the following table, we explore variants of InstructHOI that leverage different LFMs, including LLaVA-OV and InternVL2. Furthermore, for each LFM variant, we assess the impact of different foundation model scales. The experimental results demonstrate that InstructHOI achieves SOTA performance with both LLaVA-OV and InternVL2. InternVL2 demonstrates slightly better performance, which could be attributed to its strong foundational and visual perception capabilities. Additionally, as the scale of the underlying LFM increases, InstructHOI’s performance further improves, highlighting the enhanced reasoning capabilities at larger model scales.
>
> | Methods - *LFMs*           | Foundation Models           | V-COCO |
> |--------------------------|-----------------------------|:--------:|
> | InstructHOI-$_{LLaVA-OV}$      | ViT-L Qwen2$_{0.5B}$           | 70.3   |
> | InstructHOI-$_{LLaVA-OV}$      | ViT-L Qwen2$_{7B}$             | **71.4**   |
> | InstructHOI-$_{InternVL2}$   | ViT-L Qwen2$_{0.5B}$            | 70.8   |
> | InstructHOI-$_{InternVL2}$    | ViT-L InternLM2$_{7B}$        | **72.1**   |
>
>
> --------------------------------------------------------------------------------------------------------------
>
> **Q2: The necessity of Spatial Context in CIG.**
>
> **A2:** The primary objective of integrating an offline object detector is not to extract Spatial Context (SC), but to achieve precise instance localization. Consistent with prior work [21], [58] demonstrating the importance of spatial relationships for interaction prediction, the experimental results in Table 4 (in the main paper) show that the introduction of SC token improves performance on the HICO-DET by 0.78 mAP (**1.8% relative improvement**). Moreover, when the LFM model is scaled up to 7B parameters (InternVL2-7b), the gains of SC further increase to 1.42mAP (**3.1% relative improvement**) on the HICO-DET. LFMs with greater foundational capacity demonstrate an enhanced ability to comprehend SC token and leverage it for interaction reasoning.
>
> --------------------------------------------------------------------------------------------------------------
>
> **Q3: Clarify that the Backbone in Table 1.**
>
> **A3:**  For VLM-based HOI detectors, the term "backbone" (e.g., R101+ViT-L) refers to the combination of the basic feature extractor (e.g., R101 for object detection) and the visual encoder backbone of the large foundation model (e.g., ViT-L). Additionally, to provide a clearer perspective, we also add the specific Large Foundation Model (LFMs) of different methods in Table 1, as following:
>
> | Method         | Backbone        | LFM              |
> |----------------|-----------------|------------------|
> | DiffusionHOI[65] | VQGAN+ViT-L           | Stable Diffusion\&CLIP |
> | UniHOI-l[22]    | R101+ViT-L      | BLIP-2           |
> | MP-HOI[43]      | Swin-L+ViT      | Stable Diffusion\&CLIP |
> | SICHOI[23]      | R101+ViT-L      | ChatGPT\&BLIP          |
> | InstructHOI     | R101+ViT-L      | InternVL2\&CLIP        |
>
> --------------------------------------------------------------------------------------------------------------
>
> **Q4: Inference and training cost.**
>
> **A4:** We evaluated the inference time of our model in comparison to high-performance HOI detectors, utilizing Tesla A800 GPU (or GPU with comparable performance). The results are presented in the table below, compared to existing methods, InstructHOI demonstrates notable performance improvements while maintaining comparable computational costs.
>
> | Method             | Inference time (ms) | HICO-DET |
> |--------------------|:---------------------:|:----------:|
> | ADA-CM[66]         | 286                | 38.40    |
> | DiffusionHOI[65]   | 105                 | 38.12    |
> | UniHOI[43]         | 82                  | 40.95    |
> | MP-HOI[23]         | 186                 | 44.53    |
> | InstructHOI        | 267                 | **47.68**    |
>
> We evaluated the training time of our model in comparison to high-performance HOI detectors, utilizing 8 Tesla A800 GPUs (or GPUs with comparable performance). The results are presented in the table below, specifically, the 7.2 hour training time refers to the HOI-SFT time for the MLLMs, while the 11.1 hours represents the second-stage training time for the full model’s component.
>
> | Method             | Training time (h) | HICO-DET |
> |--------------------|:----------------------:|:----------:|
> | HOICLIP[44]        | 26.4                 | 34.69    |
> | DiffusionHOI[65]   | 17.2                 | 38.12    |
> | UniHOI[43]         | 18.7                 | 40.95    |
> | MP-HOI[23]         | 23.8                 | 44.53    |
> | InstructHOI        | 7.2+11.1             | **47.68**    |
>
> --------------------------------------------------------------------------------------------------------------
>
> We greatly appreciate your insightful and constructive feedback, and we hope our response could addresses your concerns. If you have any further questions, please don’t hesitate to share them with us.
>
> In the revised version of the paper. we will provide a clearer explanation of the backbone component and the role of the spatial context token, as well as include more details on the inference and training time.

---

### Official Review · Reviewer_oDfg · 2025-06-30

**Clarity:** 3
**Significance:** 3
**Originality:** 4
**Rating:** 6
**Confidence:** 5

**Summary:**

In this paper, the authors address the reasoning capabilities of the foundation model in HOI task. Different from previous methods which directly using the pretrained LFM. A large HOI image-text pair dataset is used to fine-tune the LFM by LoRA at first. Then, the LFM is adopted to predict the interaction of the H-O pair with the resoning token from the proposed Context-Aware Instruction Genrator (CIG) and Interest Token Selector (ITS). The CIG translates the HOI visual embedding into linguistic space and the ITS selects the interaction-relevant image tokens from the visual features. The extensive experiments demonstrate the effectiveness of the proposed method in improving the performance of HOI detection.

**Questions:**

1. In Line 235, the cosine similarity operation is calculated between two sets of texts. What model is used to extract the text embeddings? E.g. CLIP, BLIP, or other text encoders.

2. With the object detector and the LFM, the model already has the ability to predict the interaction probability $S_2$ of the H-O pair. The LFM also uses both of the appearance and the H-O pair embeddings, is it necessary to use the Visual Interaction Decoder?

3. The interaction-relevance matrix $M_ir$ in ITS used to filter the image token of local features, however, building the connection between the visual features from the object detector and the LFM is implicit, the matrix may not be accurate to focus on the interaction regions. Why not directly use the predicted bounding boxes to filter the image token?

4. As mentioned in weaknesses, the 140K dataset used to fine-tune the LFM may contain the HOI categories in zero-shot settings, are these related categories' data removed from the fine-tuning dataset? If not, the comparison with the previous methods on the zero-shot setting is not fair.

**Ethical Concerns:**

["NO or VERY MINOR ethics concerns only"]

**Final Justification:**

The paper proposed a novel method to enhance the reasoning capabilities of the foundation model in HOI task and achieved state-of-the-art performance on the HOI benchmark datasets. The manuscript is clearly presented with well-structured sections and figures. This paper addresses a significant problem in the HOI task, the fine-tuning strategy of the HOI model incorporating with the LFM is innovative and is important for the community.

Thank you to the authors for the detailed and thorough response. Your reply has sufficiently addressed my concerns. Given the inspiration this work may provide for future research, I am willing to increase my score.

**Limitations:**

Yes.

**Paper Formatting Concerns:**

No formatting issues.

**Quality:**

3

**Strengths And Weaknesses:**

Strengths:

The paper proposed a novel method to enhance the reasoning capabilities of the foundation model in HOI task and achieved state-of-the-art performance on the HOI benchmark datasets. The manuscript is clearly presented with well-structured sections and figures. This paper addresses a significant problem in the HOI task, the fine-tuning strategy of the HOI model incorporating with the LFM is innovative and is important for the community.

Weaknesses:

As the LFM is fine-tuned on a 140K HOI image-text pair dataset, which contains whole HOI categories, the comparison with the previous methods on the zero-shot setting is not fair. While using the pretrained LFM, the improvement is limited. The proposed method enhances the reasoning capabilities during the fine-tuning stage of the LFM, however, the prediction is still based on the text labels and an interaction decoder is still needed to predict the interaction probability, thus, the interaction representation capabilities of the LFM is still limited.

---

> ### Author Rebuttal · Authors · 2025-07-31
>
> Dear reviewer oDfg,
>
> First and foremost, we would like to express our sincere gratitude for your detailed review and valuable feedback. We truly appreciate your recognition of our method, e.g., "novel method", "addresse a significant problem in the HOI task" and "innovative and important for the community".
>
> Below, we provide a comprehensive response to each of your concerns (please note: the metrics for HICO-DET in the table below are reported under the "default full setting," while those for V-COCO are $AP_{role}^{\\#1}$; the references maintain their original numbering from the main paper):
>
> --------------------------------------------------------------------------------------------------------------
>
> **Q1: Text embeddings model.**
>
> **A1:** Following prior work, we employ CLIP’s text encoder to embed both the generated text and the textual labels, and then compute their similarity for interaction prediction.
>
> --------------------------------------------------------------------------------------------------------------
>
> **Q2: Necessity of Visual Interaction Decoder.**
>
> **A2:** We conduct an ablation study by using only the LLM’s outputs (S2) for interaction prediction on the HICO-DET. As shown in the table below, relying solely on S2 results in a 5.09 mAP drop in the default full setting compared to the combined "S1 + S2" approach (where S1 represents interaction scores generated by the Visual Interaction Decoder). These experimental results underscore the critical role of the Visual Interaction Decoder: by fusing the visual interaction encoder (S1) with MLLM reasoning (S2), InstructHOI could effectively preserves the generalization ability of the MLLM and mitigate hallucinations, leading to superior overall performance.
>
> | Interaction Score | HICO-DET |
> |-----------------|:--------:|
> | S2                |  40.86   |
> | S1 + S2           |  **45.95**   |
>
> --------------------------------------------------------------------------------------------------------------
>
> **Q3: Directly use the predicted bounding boxes to filter the image token.**
>
> **A3:**  We conducted an experiment in which we directly used the predicted bounding boxes to filter the image tokens (region-overlap strategy). As shown in the table below, this region-overlap strategy results in a 1.22 mAP performance drop compared to the Interest Token Selector (ITS). **Directly using the detected HOI region as the interaction region may neglect discriminative features and introduce irrelevant information**.
>
> Although the connection between the visual features extracted from the object detector and the LFM is implicit, the instruction projector (Sec. 3.3) could aligns these two visual feature spaces via feature mapping. ITS evaluates the interaction relevance of each local patch for each human-object pair, thereby more effectively selecting the interaction-related region.
>
> | Token Selection Strategy  | HICO-DET |
> |---------------------------|:---------:|
> | No Selection              |    45.90 |
> | Region-overlap Strategy       |    46.46 |
> | Interest Token Selector   |    **47.68** |
>
> --------------------------------------------------------------------------------------------------------------
>
> **Q4: Removal of Data from Related Categories.**
>
> **A4:**  To ensure a fair zero-shot evaluation, we excluded all training data related to the held-out HOIs from the fine-tuning dataset, thereby preventing any potential data leakage. For instance, when "talk on a cell phone" (in HICO-DET) is considered an unseen HOI, we also exclude related HOI categories from the fine-tuning dataset, e.g., "phone" in SWIG-HOI and "talk on phone" in V-COCO.
>
> --------------------------------------------------------------------------------------------------------------
>
> We truly value your insightful and constructive suggestions, and we hope our response could addresse your concerns. If you have any further questions or feedback, please do not hesitate to share them with us.
>
> In the revised version of the paper, we will include a clearer explanation of the text embedding model, ablation experiments on the Visual Interaction Decoder, experiments on local image token filtering with detected HOI regions, and a more detailed explanation of data removal.

---

> > ### Comment · Reviewer_oDfg · 2025-08-08
> >
> > Thank you to the authors for the detailed and thorough response. Your reply has sufficiently addressed my concerns. I would also like to encourage the authors to release reproducible code for the benefit of the research community. Given the inspiration this work may provide for future research, I am willing to increase my score.

---

> > > ### Author Response · Authors · 2025-08-08
> > >
> > > Thank you for taking the time to review our response and provide valuable feedback.
> > >
> > > We are pleased to hear that your concerns have been sufficiently addressed and that the score has improved. We will release the code and data. Please do not hesitate to reach out if you have any further questions.

---

### Official Review · Reviewer_hkno · 2025-07-03

**Clarity:** 3
**Significance:** 3
**Originality:** 3
**Rating:** 4
**Confidence:** 3

**Summary:**

The paper introduces InstructHOI, a novel architecture that leverages context-aware instructions to guide large foundation models (LFMs) in performing multimodal reasoning for human-object interaction (HOI) detection. A Context-aware Instruction Generator (CIG) is developed to fuse visual interactive context (appearance/spatial features) with linguistic instructions, providing pair-level reasoning guidance that outperforms language-only instructions. An Interest Token Selector (ITS) dynamically filters and reorganizes image tokens based on context-aware instructions, effectively aligning the reasoning process with relevant interaction regions while reducing noise. Extensive experiments demonstrate state-of-the-art results on HICO-DET and V-COCO datasets.

**Questions:**

Please see [Weaknesses] section.

**Ethical Concerns:**

["NO or VERY MINOR ethics concerns only"]

**Final Justification:**

Thanks for the author's detailed response. Most of my concerns have been addressed. Given that my initial score was already leaning positive, I will keep the score unchanged.

**Limitations:**

yes

**Paper Formatting Concerns:**

NIL

**Quality:**

3

**Strengths And Weaknesses:**

[Strengths]

1. The paper is well-written and easy to follow.

2. Clear motivation and novel architecture design. The paper identifies the limitation of prior LFM-based HOI approaches, which focus mostly on static prompt-based knowledge transfer and largely ignore reasoning capabilities. And it introduces InstructHOI to leverage context-aware instructions for guiding large foundation models (LFMs) in human-object interaction (HOI) detection.

3. The proposed CIG combines visual-human-object context (appearance and spatial features) with linguistic instructions for more contextually aware reasoning, which is more advanced than conventional prompts. The Interest Token Selector (ITS) dynamically filters and reorganizes image tokens based on interaction relevance, effectively focusing the model's attention on crucial human-object regions. This spatial adaptation mechanism significantly improves reasoning accuracy.

4. SoTA performance. Extensive experiments demonstrate state-of-the-art results on HICO-DET and V-COCO datasets.

[Weaknesses]

1. Lack of efficiency metrics. The paper only compares the backbone networks of different LFM-based HOI methods (Table 1 in the main paper) and trainable parameters (Table 1 in the supplementary materials), while training and inference costs are also important. It is suggested that the authors provide in Table 1 of the main text the specific LFM used by different methods and their corresponding parameter counts, the amount of training data used, average inference time, FLOPs, and other metrics.

2. Lack of visualizations. The proposed Interest Token Selector (ITS) module is interesting, and visualizations could more clearly validate its effectiveness. The authors only provide one visualization result of ITS in the supplementary materials, which raises questions about whether this is coincidental or whether there are bad cases. It is suggested that the authors provide more visualization results.

3. Applying LFMs to the HOI domain is a challenging topic. This paper collected 5 existing image-only HOI datasets for HOI-domain fine-tuning, which is used for subsequent Multi-Modal Reasoning (MMR). Given the powerful visual understanding capabilities of multimodal large models, is LoRA fine-tuning alone sufficient to perform excellently on HOI datasets? It is suggested that the authors provide this result as an additional baseline.

---

> ### Author Rebuttal · Authors · 2025-07-31
>
> Dear reviewer hkno,
>
> First and foremost, we extend our deepest gratitude for your thorough review and insightful feedback, as well as your positive recognition of our method, e.g.,  "clear motivation", "novel architecture design" and "SoTA performance".
>
> Herein, we provide a detailed response to each of your concerns (note: the metrics for HICO-DET in the following table are reported under the "default full setting", while those for V-COCO are  $AP_{role}^{\\#1}$;  the references maintain their original numbering from the main paper):
>
> --------------------------------------------------------------------------------------------------------------
>
> **Q1: Efficiency metrics in Table 1.**
>
> **A1:** Following prior work, we add the most commonly used metrics ("Large Foundation Model" (LFM), "Training Data" (TD), "Full Parameters" (FP), "Trainable Parameters" (TP), "Inference Time" (IT), and "Training Time" (TT))  in Table 1, to provide a clearer efficiency comparison, as shown below. All measurements are obtained using Tesla A800 GPUs (or GPUs with comparable performance).
>
> | Method       | LFM           | TD(K)    | FP(B)      | TP(M)     | IT(ms)    | TT(h)        | HICO-DET | V-COCO |
> |--------------|---------------|:--------:|:---------:|:--------:|:-------:|-----------|:----------:|:--------:|
> | DiffusionHOI[65] | Stable Diffusion\&CLIP | 48    | 1.02   | 27.6  | 105 | 17.2     | 38.12    | 66.8   |
> | UniHOI-l[22]       | BLIP-2        | 48    | 1.14   | 52.3  | 82  | 18.7  | 40.95    | 68.1   |
> | MP-HOI[43]       | Stable Diffusion\&CLIP | 286   | 1.05   | 41.5  | 196 | 23.8     | 44.53    | 66.2   |
> | SICHOI[23]       | ChatGPT\&BLIP       | 48    | -       | -      | -     | -         | 45.04    | 71.1   |
> | InstructHOI  | InternVL2\&CLIP     | 140   | 1.13  | 42.3  | 267 | (7.2+11.1)| **47.68**    | **72.4**   |
>
> Compared to existing methods, InstructHOI demonstrates notable performance improvements while maintaining comparable model parameters (1.13B full parameters and 42.3M trainable parameters) and computational costs (267ms inference time and a total training time of 18.3 hours). Specifically, the 7.2 hour training time refers to the HOI-SFT time for the MLLMs, while the 11.1 hours represents the second-stage training time for the full model’s component.
>
> --------------------------------------------------------------------------------------------------------------
>
> **Q2: Lack of visualizations.**
>
> **A2:**  Thank you for pointing this out. Due to rebuttal guidelines, we are unable to upload images here. As for Interest Token Selector (ITS) module, we ensure that additional visualizations will be introduced in the revised version to more clearly validate  the effectiveness. Other visualizations—covering Interaction Prediction in Two Branches, Failure Case Analysis, and In-the-Wild HOI Detection—have been provided in the supplementary materials.
>
> Additionally, to quantitatively validate the effectiveness of ITS module, we have conducted an ablation experiment in Table 3 (main paper). The results demonstrate a 0.97 mAP increase on HICO-DET (a relative gain of 2.2 %) and a 0.6 mAP increase on V-COCO (a relative gain of 0.9 %).
>
> --------------------------------------------------------------------------------------------------------------
>
> **Q3: Directly use LoRA-finetuned MLLM for interaction prediction.**
>
> **A3:**  We introduce an additional baseline in which the LoRA-finetuned MLLM is directly employed for HOI interaction recognition, referred to as Multi-Modal Reasoning (MMR). As shown in the table below, MMR alone results in a 5.09 mAP drop on the HICO-DET compared to InstructHOI. The standalone MMR may be prone to hallucinations in complex and ambiguous scenarios. In contrast, integrating the visual interaction decoder (S1) with MMR (S2) for HOI detection effectively maintains the generalization capability of the MLLM while mitigating hallucinations, leading to a significant improvement in overall performance.
>
> | Prediction Method | HICO-DET |
> |-------------------|:----------:|
> | MMR               | 40.86    |
> | InstructHOI       | **45.95**  |
>
> --------------------------------------------------------------------------------------------------------------
>
> We sincerely appreciate your valuable and constructive suggestions, and we hope that our response could addresses your concerns. Please feel free to provide any further feedback if you have additional questions.
>
> In addition, we will incorporate your suggested revisions into the future version of the paper, including: adding efficiency metrics in Table 1, providing visual results for ITS, and including additional baseline experiments on directly using LoRA-finetuned MLLMs for interaction prediction. These revisions will significantly enhance the quality of the paper, allowing readers to gain a more comprehensive understanding and evaluation of our work.

---

> > ### Comment · Reviewer_hkno · 2025-08-07
> >
> > Thank you for the author's detailed response. Most of my concerns have been addressed. Given that my initial score was already leaning positive, I will keep the score unchanged.

---

> > > ### Author Response · Authors · 2025-08-07
> > >
> > > Thank you for taking the time to review our response and for your valuable feedback. We're glad to hear that most of your concerns have been addressed. Please feel free to reach out if you have any further questions.

---

### Official Review · Reviewer_cgtz · 2025-07-22

**Clarity:** 3
**Significance:** 3
**Originality:** 2
**Rating:** 5
**Confidence:** 4

**Summary:**

This paper proposes a new model for Human-Object Interaction (HOI) detection. Specifically, the authors propose a two-branch model in which the first branch includes three modules: Context-aware Instruction Generator (CIG), Interest Token Selector (ITS), and Multi-Modal Reasoning (MMR). CIG works by leveraging an off-the-shelf object detector (DETR) to detect both humans and the objects, use the detected HOI region to compute spatial and appearance features, and then fuse the features with the textual instructions (e.g. "Reason the interaction between human and objects") to produce a "Context-aware instruction". Then, ITS takes the output from CIG to predict which local image split best correlates with the HOI interactions and use that to guide the local image features to be fed into the MMR module (i.e. the LLM). The MMR will then decode the text outputs describing the human-object interactions.

The second branch is somewhat standard in that it takes in the input image, encode the image with the vision encoder, then it has a Visual Interaction Decoder that takes in the appearance features from the detected HOI region as well as the global vision features to predict the HOI label out from a vocabulary.

The authors have tested the proposed method on two prominent HOI datasets HICO-DET and V-COCO, and demonstrated that the proposed model has consistently improved over prior state-of-the-arts.

**Questions:**

- Have you experimented with full finetuning for the language model? Wonder how that works compared to LORA tuning adopted in the paper
- L152, why freeze the image encoder? As several papers (e.g. Cambrian paper by Tong et al) suggested the benefits of joint tuning for image encoders.
- For training the Multi-modal Reasoning module, have you considered directly task it to predict labels in L_t's taxonomy? What's the benefits of computing textual similarity between L_t and L_p?

**Ethical Concerns:**

["NO or VERY MINOR ethics concerns only"]

**Final Justification:**

I thank the authors for their efforts in the rebuttal. The authors have addressed most of my concerns and questions, I'm overall happy with the quality of the paper and the rebuttal consolidated my position view on the contribution of this work. Thus I will keep my rating to advocate accept the paper.

**Limitations:**

Yes

**Quality:**

3

**Strengths And Weaknesses:**

+ The paper is mostly well-written. Despite the many modules and branches in the proposed model, the authors have generally done a good job describing the proposed method. As for the description of the experiments, the texts are also clear and to the point.
+ The proposed method, despite the somewhat complicated design, overall looks reasonable. I did not catch any obvious technical errors.
+ The proposed method has been shown to consistently improve over prior state-of-the-arts on Human-Object-Interaction, across two prominent benchmarks.
+ The ablation experiments presented in the Experiments section dissect different components of the proposed method, which is particularly important given the multi-stage nature of the method.
- Table 1, in addition to the R50 + ViT-L, the proposed method also leverages a ViT for visual interaction decoder, and a LLM which should be taken into account when comparing different models. Can you provide the full parameter counts of the model including all modules? It would be good to have full parameter size comparison in Table 1.
- The proposed model consists of a complicated pipeline involving several models (DETR for object detection, InternVL2 image encoder and LLM, ViT for visual interaction decoder), it's important to establish a comparison to end-to-end prediction using recent VLM models with decent sizes (e.g. Qwen2.5-VL models of 3B/7B sizes). Note this is different to providing justification to the effectiveness of each individual component as is done in the current ablation studies.
- Regarding the Interest Token Selector (ITS), an important ablation that's missing is to compare to just selecting the local split that overlaps the most with the detected HOI regions.
- Eq 4, what's the shape of T_l? It's not clear from the texts how \hat{T}_{in} and T_l are concatenated
- Eq 8, we need an ablation to test the accuracy of just using the LLM outputs (S2)

---

> ### Author Rebuttal · Authors · 2025-07-30
>
> Dear reviewer cgtz,
>
> First and foremost, we sincerely appreciate your thorough review and valuable feedback. Your recognition of our method is greatly appreciated, e.g., "well-written", "reasonable design" and "ablation experiments given the multi-stage nature of the method".
>
> Below, we provide detailed responses to each of your questions (note: the metrics for HICO-DET in the following table are reported under the "default full setting", while those for V-COCO are  $AP_{role}^{\\#1}$; the references maintain their original numbering from the main paper):
>
> --------------------------------------------------------------------------------------------------------------
>
> **Q1: Full parameter size comparison in Table 1.**
>
> **A1:** For a clearer comparison, we add the "Large Foundation Model” (LFM) and the “Full Parameters”(FP) and “Trainable Parameters” (TP) in Table 1, as shown below. Compared to existing methods, InstructHOI achieves substantial performance gains with comparable number of trainable parameters (42.3M) and full parameters (1.13B).
>
> | Method        | LFM             |    FP(B)    |    TP(M)   |   HICO-DET   |   V-COCO    |
> |---------------|-----------------|:----------:|:-------------------------:|:--------------:|:------------------------------:|
> | DiffusionHOI[65]  | Stable Diffusion\&CLIP |  1.02   |  27.6   |    38.12     |    66.8     |
> | UniHOI-l[22]      | BLIP-2          |  1.14   |  52.3   |    40.95     |    68.1     |
> | MP-HOI[43]        | Stable Diffusion\&CLIP|  1.05   |  41.5   |    44.53     |    66.2     |
> | SICHOI[23]        | ChatGPT\&BLIP        |    -     |    -     |    45.04     |    71.1     |
> | InstructHOI  | InternVL2\&CLIP       |  1.13   |  42.3   |    **47.68**     |    **72.4**     |
>
> Thank you for your valuable suggestions, we will revise Table 1 in future versions for clearer comparison.
>
> --------------------------------------------------------------------------------------------------------------
>
> **Q2: End-to-end prediction using VLM models.**
>
> **A2:** We develope an end-to-end HOI detector using Qwen2.5VL-7B for comparison. As shown in the table below, the pretrained **Qwen2.5VL-7B** achieves a **24.72 mAP** on HICO-DET (with the default full setting), while the fine-tuned **Qwen2.5VL-7B-SFT** (fine-tuned on the HOI dataset) improves this performance to **32.64 mAP**. However, both variants are significantly outperformed by InstructHOI (only equipped with a 1B-parameter MLLM). This performance gap can primarily be attributed to the limited grounding precision and reduced interaction-prediction capabilities of MLLMs models within ambiguous and complex scenarios.
>
> | Model               | HICO-DET |
> |---------------------|:----------:|
> | Qwen2.5VL-7B        | 24.72    |
> | Qwen2.5VL-7B-SFT    | 32.64    |
> | InstructHOI         | **45.95**    |
>
> --------------------------------------------------------------------------------------------------------------
>
> **Q3: Select the local split with the detected HOI regions.**
>
> **A3:** We conduct an experiment in which we select the local split that overlaps the most with the detected HOI regions (Overlap-based Strategy). As shown in the table below, this overlap-based strategy results in a 1.22 mAP performance drop compared to the Interest Token Selector (ITS). Directly using the detected HOI region as the interaction region may **neglect discriminative features and introduce irrelevant information**. In contrast, ITS evaluates the interaction relevance of local images for each human-object pair based on reasoning instructions, allowing it to more effectively select the region most closely associated with the interaction.
>
> | Token Selection Strategy | HICO-DET |
> |--------------------------|:----------:|
> | No Selection             | 45.90    |
> | Overlap-based Strategy      | 46.46    |
> | Interest Token Selector  |  **47.68**    |
>
> --------------------------------------------------------------------------------------------------------------
>
> **Q4: The shape of $T_l$.**
>
> **A4:** In Eq. (4), $ T_l $ denotes the discretized local image token sequence generated with the dynamic image encoding strategy of InternVL2, with a shape of $ N_l \times N_s \times d $ (where $ N_l $ represents the number of local images and $ N_s $ represents the number of tokens per local image). $ \overline{{T}}_{in} $ refers to the representative instruction tokens, with a shape of $ N_p \times n \times d $. Consequently, the concatenated feature has a shape of $ N_p \times N_l \times (N_s + n) \times d $.
>
> --------------------------------------------------------------------------------------------------------------
>
> **Q5: Ablation experiment using the LLM outputs (S2) in Eq. (8).**
>
> **A5:** We conduct an ablation study by using only the LLM’s outputs (S2) for interaction prediction (in E.q.(8)) on the HICO-DET. As shown in the table below, this configuration suffers a 5.09 mAP decline in the default full setting compared to the combined "S1 + S2" approach. In contrast, integrating the visual interaction decoder (S1) with MLLM reasoning (S2) for HOI detection effectively preserves the generalization ability of the MLLM while mitigating hallucinations, leading to a noticeable improvement in overall performance.
>
> | Interaction Score | HICO-DET |
> |-------------------|:----------:|
> | S2                | 40.86    |
> | S1+S2             | **45.95**    |
>
> --------------------------------------------------------------------------------------------------------------
>
> **Q6: Full finetuning vs LoRA finetuning.**
>
> **A6:** We conduct a comparative analysis of full fine-tuning versus LoRA fine-tuning strategies for HOI detection (see the table below). The results show that LoRA fine-tuning outperforms full fine-tuning, yielding 45.95 mAP versus 43.54 mAP. As a **lightweight adaptation method**, LoRA more effectively **preserves LFM’s knowledge learned in pre-trained stage and maintains the  generalization capability**, thus enhaning the interaction reasoning capabilities in complex scenarios.
>
> | Finetuning Strategy | HICO-DET |
> |---------------------|:----------:|
> | Full Finetuning     | 43.54    |
> | LoRA Finetuning     | **45.95**    |
>
> --------------------------------------------------------------------------------------------------------------
>
> **Q7: Freeze or unfreeze the image encoder.**
>
> **A7:** We conduct an ablation experiment by jointly training the LLM, the image encoder and the V-L MLP. The experimental results, presented in the table below, show that training only the LLM yields a relative advantage, with a 1.33mAP improvement in mAP. In the InstructHOI, we fine-tune the language model of LFMs to better understand the reconstructed HOI-specific token sequences. For relatively small-scale datasets and single-domain fine-tuning, freezing the image encoder while training only the language model may help prevent catastrophic forgetting and preserve the generalization ability of pre-trained LFMs.
>
> | Training Strategy | HICO-DET |
> |-------------------|:----------:|
> | Unfreeze          | 44.62    |
> | Freeze            | **45.95**    |
>
> --------------------------------------------------------------------------------------------------------------
>
> **Q8: Task interaction reasoning to predict labels in L_t's taxonomy?**
>
> **A8:** Computing textual similarity provides a more flexible approach to open-world interaction recognition. For instance, detecting unseen or novel interaction categories can be achieved through cosine similarity matching between the generated texts from MLLMs and the textual labels of unseen or novel categories. In contrast, directly assigning interaction reasoning to predict labels within the taxonomy of L_t's requires the design of specific classification heads for predefined interaction categories. However, when new interaction categories are introduced, the model structure needs to be modified and retrained.
>
> --------------------------------------------------------------------------------------------------------------
>
> Overall, we sincerely appreciate your valuable and constructive suggestions, and hope our responses could addresse your concerns. Please don’t hesitate to share any further feedback or questions.
>
> In addition, we will incorporate the suggested revisions into the future version of the paper, e.g., the parameter count in Table 1, end-to-end prediction using VLM, local image token filtering with overlap-based strategy, a clearer explanation of E.q. (4), the ablation experiment on S1 and S2 in E.q. (8), and so on. These revisions will significantly enhance the quality of the paper, enabling readers to gain a more comprehensive understanding and evaluation of our work.

---

### Note · Authors · 2025-08-13

We sincerely appreciate the reviewers and Area Chairs for their tremendous efforts in reviewing this paper. The common questions raised by the reviewers include: (1) **efficiency evaluation** (hkno, jzDg); (2) **token selection in the Interest Token Selector (ITS)** (cgtz, oDfg); and (3) **interaction prediction with MLLM** (cgtz, hkno). The insightful feedback from the reviewers has inspired us to strengthen the contributions and depth of this research through new experimental evidence.

**Regarding (1) efficiency evaluation**, we added common metrics like "model size," "training data," "parameters," and "inference time" for clearer comparison. Compared to existing methods, our method demonstrates significant performance improvements while maintaining comparable model parameters and computational costs.

**For (2) token selection in ITS**, we experimented by selecting the local split that most overlaps with detected HOI regions. This overlap-based strategy resulted in a **1.22 mAP** performance drop compared to ITS. Directly using the detected HOI region may overlook discriminative features and introduce irrelevant information, whereas ITS, by evaluating interaction relevance, enables more effective region selection.

**For (3) interaction prediction with MLLM**, an ablation study showed a **5.09 mAP** drop when using MLLM alone. This highlights the importance of the Visual Interaction Decoder, which, by integrating with MLLM reasoning, improves generalization and reduces hallucinations, leading to better overall performance.

We hope the response and the rebuttal can effectively address the reviewers' concerns. We also hope that our work can contribute to further HOI detection research and provide meaningful value to the community.

---

### Decision · Program_Chairs · 2025-09-17

**Decision:**

Accept (spotlight)

**Comment:**

The paper presents a novel 2 branch system for the task of HOI detection. The first branch is a more standard vision encoder -> HOI decoder, which has been shown many times to be effective. The second branch is specially designed for the HOI task and imbues the model the ability to take advantage of context-aware instructions that encourage better reasoning about the interactions taking place in the image.  This idea makes a lot of sense and the proposed system leverages current llm/vlm tools well to achieve this better reasoning. Importantly the authors show that using this method they can train on a large corpus of datasets (combining HOI datasets) to show generalization as well as just the specific train split for the test set to show fairness to other methods. The authors are careful about removing HOIs that are present in any test set to make sure their method is reasoning across unseen interaction pairs not just memorizing.

The combination of logical and novel system with strong, consistent improvements to SOTA warrant selection as an spotlight. The method could be a standard baseline for HOI reasoning going forward.

All reviewers agree the paper should be accepted and the AC agrees. The paper is a strong candidate for an spotlight.